evolution

anisogamy, size dimorphism, queen, male, superorganism

**Author for correspondence:**
Jussi Lehtonen
e-mail: jussi.lehtonen@iki.fi

# Superorganismal anisogamy: queen–male dimorphism in eusocial insects

Jussi Lehtonen[1] and Heikki Helanterä[2]

[1]Faculty of Science, School of Life and Environmental Sciences, The University of Sydney, Sydney, New South Wales 2006, Australia
[2]Ecology and Genetics Research Unit, University of Oulu, Oulu, Finland

JL, 0000-0001-5260-1041; HH, 0000-0002-6468-5956

Colonies of insects such as ants and honeybees are commonly viewed as 'superorganisms', with division of labour between reproductive 'germline-like' queens and males and 'somatic' workers. On this view, properties of the superorganismal colony are comparable with those of solitary organisms to such an extent that the colony itself can be viewed as a unit analogous to an organism. Thus, the concept of a superorganism can be useful as a guide to thinking about life history and allocation traits of colonies as a whole. A pattern that seems to reoccur in insects with superorganismal societies is size dimorphism between queens and males, where queens tend to be larger than males. It has been proposed that this is analogous to the phenomenon of anisogamy at the level of gametes in organisms with separate sexes; more specifically, it is suggested that this caste dimorphism may have evolved via similar selection pressures as gamete dimorphism arises in the 'gamete competition' theory for the evolution of anisogamy. In this analogy, queens are analogous to female gametes, males are analogous to male gametes, and colony survival is analogous to zygote survival in gamete competition theory. Here, we explore if this question can be taken beyond an analogy, and whether a mathematical model at the superorganism level, analogous to gamete competition at the organism level, may explain the caste dimorphism seen in superorganismal insects. We find that the central theoretical idea holds, but that there are also significant differences between the way this generalized 'propagule competition' theory operates at the levels of solitary organisms and superorganisms. In particular, we find that the theory can explain superorganismal caste dimorphism, but compared with anisogamy evolution, a central coevolutionary link is broken, making the requirements for the theory to work less stringent than those found for the evolution of anisogamy.

## 1. Introduction

Colonies of social insects such as ants have been likened to multicellular organisms ('superorganisms') for well over a century [1–3]. This analogy rests on the definition of a superorganism as a society with irreversible morphological specialization and mutual interdependence of reproductive and non-reproductive castes (for a review on the history of the usage and interpretation of the term, see [4]). Queens and males are the reproductive tissues of the colony organism, and workers the somatic tissue, sometimes further diverged into different subcastes, specialized into defence, foraging and brood care, for example. The evolution of such 'superorganismal' societies is a key example of a major transition in evolution [4–6].

Despite the popularity of the superorganism concept, the extent to which it has directly motivated empirical or theoretical research is arguably limited (but see [7] for a recent review). There are insightful descriptions of colony function based on hierarchical comparisons of the anatomy, physiology and regulation in

**Figure 1.** Gametic reproduction results in fitness benefits if the zygote survives. Reproduction by a queen founding a superorganismal colony results in fitness benefits (i.e. queens and males that mate to form the next generation) if the incipient colony survives, analogous to the gamete-level case. By contrast, in reproduction by solitary organisms, there is no stage in the life cycle that is analogous to a zygote or incipient colony, and a mated female can immediately reproduce. This is why there is a clear analogy between the evolution of gamete dimorphism and queen–male dimorphism in eusocial insects, but not between the evolution of gamete dimorphism and sexual dimorphism in solitary organisms.

multicellular organisms and insect societies [8,9]. However, no novel testable predictions have been emerging from such approaches, and despite the extremely diverse life histories and allocation strategies of superorganisms [10], few attempts to explain the variation with organismal, rather than social insect specific theories, have been made (but see [11] for a notable exception).

Using the organismal analogy to explain life history and allocation strategies of social insect colonies [10] requires adapting existing theoretical models to fit social insect biology, as their assumptions will typically not directly fit the colonial superorganism. In this paper, we make such an attempt and apply gamete competition models of anisogamy evolution (i.e. the divergence of gamete sizes, such as egg and sperm [12–15]) to superorganisms, motivated by the verbal analogy of Helanterä [10]. If we take a social insect colony as a superorganism rather than a society formed of individual insects, the dispersing young males and future queens are analogous to the gametes of a multicellular organism, i.e. sperm/pollen and eggs/seeds, respectively. They leave their mother colony, disperse and mate to form a zygote-like incipient colony, i.e. a mated queen (accompanied by a king in the case of termites) ready to lay eggs that will develop into workers, that form the somatic body of the colony. This 'zygotic' stage of obligatory production of non-reproductive workers is what sets apart a superorganismal life cycle from a female solitary organism who can reproduce immediately after mating. That is, there is an obligatory somatic growth stage without which reproduction is completely impossible [16] (figure 1), analogous to the development of a zygote where all fitness gains are lost if the zygote does not survive into the reproductive stage of the life cycle.

The gamete competition theory for the evolution of anisogamy aims to explain the divergence of gamete sizes into male and female gametes, and analyses the conditions under which such a divergence is expected to occur. The model assumes that there is 'fair raffle' type of competition between gametes over fertilizations, resulting in a selection pressure to produce more numerous (and hence smaller) gametes. At the same time, the developmental requirements of the zygote result in a selection pressure to produce larger gametes for provisioning the zygote. In its modern form [17], gamete competition theory suggests that gamete sizes begin to diverge once the developmental requirements of the zygote diverge far enough from the minimum size requirements of the gamete itself, as is expected to happen as multicellularity evolves. Under these conditions, disruptive selection can lead to the production of large gametes (e.g. eggs) that provide resources to the offspring, and small gametes (e.g. sperm) that 'compete' for fertilizations. Gamete competition theory remains the most widely accepted explanation for the evolution of anisogamy, and its predictions have empirical support, although exceptions are also known (see [13–15] for recent reviews).

Analogues of the basic assumptions of gamete competition models are likely to apply at the superorganism level as well. First, even if male behaviour, queen mate choice and sexual selection in social insects are still understudied (but see [18,19]), the assumptions of random mating with respect to size and a scramble-like competition for matings (analogous to fair raffle competition at the gamete level) are likely to largely apply [10]. There is little evidence of aggressive male–male competition or mate choice by queens [20]. Second, resources available to the 'zygotic' incipient colony increase its survival prospects, so that initial colony (zygote) size brings benefits of size beyond survival to mating. The more resources there are available at the very earliest stage of colony founding, the more workers can be produced rapidly without external resources. The colony foundation stage when the colony consists of a queen and few workers only is a period of high mortality due to conspecific competition [21,22], such as brood raiding in the fire ant *Solenopsis invicta* [23]. If fast growth is indeed a key determinant of colony survival, and if a queen with ample resources can lay eggs faster, the assumption of a positive relationship between initial colony resources and colony survival (analogous to a positive

**Table 1.** Model notation, variables and parameters.

| variable or parameter | notation | notes |
|---|---|---|
| total resource allocation to queens and males | $M$ | allocation to queens and males is assumed to be equal |
| resources allocated to a single queen: resident value (mutant value) | $x$ ($\hat{x}$) | determines queen size |
| resource allocation to a single male: resident value (mutant value) | $y$ ($\hat{y}$) | determines male size |
| number of queen offspring: resident value (mutant value) | $n_x$ ($\hat{n}_x$) | $n_x = \dfrac{M}{x}$ |
| number of male offspring: resident value (mutant value) | $n_y$ ($\hat{n}_y$) | $n_y = \dfrac{M}{y}$ |
| parameter controlling relationship between individual size and individual survival (analogous to gamete survival) | $\alpha$ | assumed to be the same for queens and males |
| parameter controlling relationship between queen size and colony survival (analogous to zygote survival) | $\beta$ | |
| individual survival until mating and colony founding | $g(z) = e^{-\alpha/z}$ | $z$ stands for $x$, $y$, $\hat{x}$ or $\hat{y}$ |
| colony survival | $s(z) = e^{-\beta/z}$ | $z$ stands for $x$ or $\hat{x}$, except for the multi-queen model where several queens contribute to $z$ |

relationship between zygote size and zygote survival) probably holds. One key point where the process differs at the gamete and superorganism levels is that in some species of ants, colonies are founded by multiple queens [24,25] and this may alter the queen size–colony survival relationship (see model 3).

Thus, superficially the analogous evolution of anisogamy at two different hierarchical level seems a feasible hypothesis. Furthermore, even if the empirical patterns have not been systematically analysed, at least in ants, queens seem to be consistently larger than males (see [10] and references therein). In this paper, we adapt gamete competition models of anisogamy evolution to fit the reproductive biology of superorganisms. We explore whether similar processes could underlie evolution of anisogamy at different hierarchical levels and highlight the key similarities and differences in the model predictions.

## 2. Models

Our models are influenced by several earlier models on the evolution of anisogamy, but the central paper that we have followed is that of Bulmer & Parker [17], a game theoretical update of classic theory on the evolution of anisogamy [12]. The work of Bulmer & Parker is a particularly suitable starting point for us for several reasons. It begins with the assumption of pre-existing mating types (contra [12], in which all possible pairs of gametes were able to fuse with each other), analogous to our assumption of pre-existing male and female sexes. Second, it analyses the requirements for the evolution of anisogamy and how they relate to size-specific survivorship of gametes and zygotes: one of our aims is to determine whether an equivalent of these requirements exists at the superorganism level. Third, it is arguably the most complete analysis of gamete competition models while excluding gamete limitation, which is an alternative (or complementary) pathway to the evolution of anisogamy [13]. Later analyses of anisogamy evolution have combined both gamete competition and gamete limitation in a single model (beginning with [26]), but here we exclude gamete limitation and its superorganismal equivalent mate limitation for biological reasons (it is reasonable to assume that matings of the queens

are not limited by availability of males) as well as reasons of clarity and mathematical simplicity. Thus, our primary aim is to determine whether an equivalent of gamete competition models of anisogamy evolution can drive the evolution of queen–male dimorphism in superorganismal insects.

## (a) Model 1

We begin with a modification of the first model (model $b$ in their notation) from Bulmer & Parker [17]. We focus on the salient differences between the evolution of gamete dimorphism versus queen–male dimorphism, and do not explicitly consider the role of workers or any relatedness asymmetries. We simply assume a fixed sex allocation of 1 : 1, that the size of queen and male offspring are evolving traits and that there is a positive relationship between queen size and colony survival. Notation and definitions for all models are given in table 1. The functions for offspring size-number trade-offs, for individual survival and for colony survival in table 1 follow those used by Bulmer & Parker [17] in their gamete-level model. The function used for colony survival was originally derived for survivorship of marine invertebrate eggs [27,28]. It is based on the premise that larger eggs develop more quickly and are thus under mortality risk for a shorter period of time. The derivation starting from this premise [27,28] results in the function $s(z)$ (table 1), which increases initially in an accelerating fashion until the increase eventually slows down and the function approaches a maximum value of 1 (corresponding to a maximum survival probability of 1). The parallel premise of decreased colony development time with larger queen size is reasonable for superorganisms [10], hence justifying the use of the same function here. It is not as obvious that the same function should apply to the survival of individual queens and males, and hence we follow Bulmer & Parker [17] and use the same function (denoted $g(z)$ in table 1) for individual survival (model 1), but also take the alternative approach of having a fixed minimum size (model 2).

Total allocation to male and queen offspring is assumed to be equal, but offspring numbers depend on their sizes. We assume that all queens are fertilized on their nuptial

flight, so that males are effectively competing for a fixed total fitness pot for any given queen size.

As a clear departure from the logic of the anisogamy model of Bulmer & Parker [17], in the current model, males only contribute genes and no material resources for colony or offspring development (as is the case in social Hymenoptera, where males die after mating without further contributions to colony success). In gamete evolution models, the size of both gametes can contribute material resources to offspring fitness, particularly when the two gametes are similar in size. A spermatozoon is typically assumed to contribute near-zero resources, but tiny sperm represent only one end of a continuum from isogamy (equal contribution) to extreme anisogamy. This difference removes one coevolutionary aspect of the anisogamy model.

Males face scramble competition so that each surviving male has equal chances of mating regardless of his size. These assumptions imply that the structure of the model is unaffected by the number of mating partners of a queen or male: if each male is considered analogous with a raffle ticket, then the expected winnings per male do not depend on the extent of multiple mating by the queens (although the distribution and variance of male fitness outcomes may do so). In other words, if total fitness is $w_{tot}$ and there are $N$ males competing for it, then the mean fitness per male is $w_{tot}/N$ so that the expected value of each raffle ticket is the same regardless of how the prize is distributed between them.

We assume that the male population is large and well mixed to the extent that when a mutant mother producing a slightly deviant number of male offspring appears, this initially has a negligible effect on the extent of competition faced by male offspring. In other words, although a mutant queen produces $\hat{n}_y$ sons, there are still on average $n_y$ males competing for every $n_x$ queens in the population, and in the competitive environment of any male seeking matings. We also assume that mutations affecting male and queen sizes appear rarely enough that the two types of mutations are never segregating in the population simultaneously. Therefore, mutant males never compete for mutant queens.

Now we can write the fitness of a daughter carrying a mutant allele as the product of her own survival until colony founding, and the subsequent survival of the colony

$$g(\hat{x})s(\hat{x}). \tag{2.1}$$

The fitness of a son carrying a mutant allele is

$$g(\hat{y})\frac{n_x g(x)s(x)}{n_y g(y)}. \tag{2.2}$$

The logic of equation (2.2) is as follows: in the well-mixed population, an average of $n_x g(x)s(x)$ colony fitness units are distributed over $n_y g(y)$ surviving males. Hence, the latter component of equation (2.2) is simply the average amount of fitness per male surviving to mate. The focal mutant male survives to mate with probability $g(\hat{y})$, yielding an average fitness per mutant male of $g(\hat{y})\frac{n_x g(x)s(x)}{n_y g(y)}$.

Hence, the total fitness a focal mother gains via all mutant daughters is

$$\hat{w}_x = \hat{n}_x g(\hat{x})s(\hat{x}), \tag{2.3}$$

and via all mutant sons

$$\hat{w}_y = \hat{n}_y g(\hat{y})\frac{n_x g(x)s(x)}{n_y g(y)}. \tag{2.4}$$

Substituting the functions for $n$, $g$ and $s$ (table 1), we can compute the direction of selection for each as

$$\frac{1}{w_x}\frac{\partial \hat{w}_x}{\partial \hat{x}}\bigg|_{\hat{x}=x} = \left(\frac{\hat{n}_x'}{n_x} + \frac{g'(\hat{x})}{g(x)} + \frac{s'(\hat{x})}{s(x)}\right)\bigg|_{\hat{x}=x} = -\frac{1}{x} + \frac{\alpha}{x^2} + \frac{\beta}{x^2} \tag{2.5}$$

and

$$\frac{1}{w_y}\frac{\partial \hat{w}_y}{\partial \hat{y}}\bigg|_{\hat{y}=y} = \left(\frac{\hat{n}_y'}{n_y} + \frac{g'(\hat{y})}{g(y)}\right)\bigg|_{\hat{y}=y} = -\frac{1}{y} + \frac{\alpha}{y^2}. \tag{2.6}$$

The method used in equations (2.5) and (2.6) is that of evolutionary game theory [29], which is in the context of this model effectively equivalent to adaptive dynamics (i.e. the equations are of similar form in both methodologies; see [30]). In words, equations (2.5) and (2.6) amount to estimating selection on mutants that deviate from the prevalent resident size by a small amount. Clearly, there is an asymmetry in these equations that is not present in a typical anisogamy model (e.g. [17]). The equations of selection in anisogamy models are typically symmetrical, and either gamete type can become the larger or smaller one. In equations (2.5) and (2.6), selection on queen size includes an additional component $\beta/x^2$ that is missing from selection on male size, and it is thus predetermined that queens become larger than males. The candidate evolutionarily stable strategies ($x^*$ and $y^*$) for male and queen size can be solved by setting both equations equal to zero and solving for $x$ and $y$ (i.e. finding values for queen and male sizes where selection vanishes). A straightforward calculation yields

$$x^* = \alpha + \beta \tag{2.7}$$

and

$$y^* = \alpha. \tag{2.8}$$

These equations show that under the model assumptions, queens are expected to be larger than males. Stability analysis (electronic supplementary material, appendix) shows that $x^*$ and $y^*$ are convergence stable and evolutionarily stable (see ch. 12 of [31]).

It is worth elaborating once more on why the result is different from that of the anisogamy model of Bulmer & Parker [17]. The reason is that in the anisogamy model, both gametes have potential to provision the zygote and thus influence its survival, and the resource provisioning by both gametes can be substantial in isogamous and near-isogamous reproductive systems. In the current model, males play no part in colony survival after fertilizing a queen. As the male gamete size does not enter the zygote's survival function, there is no feedback between male and queen sizes, and hence no coevolution. Mathematically speaking, where 'zygote' fitness in the current model is $s(x)$, in an anisogamy model, it would be $s(x + y)$, hence linking the fitnesses of the two gamete types in a relatively more complex coevolutionary process. In the current model, both classes reach their evolutionarily stable sizes independently. For principally the same reason, sizes of males and queens always diverge in the current model as long as $\beta > 0$, whereas in the anisogamy model of Bulmer & Parker [17], they only diverge under the condition $\beta > 4\alpha$ (that is, when the survival requirements of the zygote have sufficiently diverged from those of the gamete, as is expected when multicellularity develops).

## (b) Model 2

Now the analysis is repeated adapting model $d$ from Bulmer & Parker [17]. In their model $d$, the authors assume that gamete survival is dependent on size in a stepwise fashion, so that below a threshold size $\delta$, gamete survival equals 0, and above the limit, it equals 1. In our colony-level model, this implies that fitness via both queen and male offspring is 0 below that limit, while above the limit, they are

$$s(\hat{x}) \tag{2.9}$$

for queen offspring and

$$\frac{n_x s(x)}{n_y}. \tag{2.10}$$

for male offspring. Equations (2.9) and (2.10) are equivalent to equations (2.1) and (2.2) with the gamete survival functions ($g$) set equal to 1.

For viable sizes ($>\delta$), total fitness a focal mother gains via all mutant daughters is

$$\hat{w}_x = \hat{n}_x s(\hat{x}) \tag{2.11}$$

and via mutant sons

$$\hat{w}_y = \hat{n}_y \frac{n_x s(x)}{n_y}. \tag{2.12}$$

Substituting $n$ and $s$ (table 1), we can compute the direction of selection for each as

$$\frac{1}{w_x} \frac{\partial \hat{w}_x}{\partial \hat{x}}\bigg|_{\hat{x}=x} = \left( \frac{\hat{n}'_x}{n_x} + \frac{s'(\hat{x})}{s(x)} \right)\bigg|_{\hat{x}=x} = -\frac{1}{x} + \frac{\beta}{x^2} \tag{2.13}$$

and

$$\frac{1}{w_y} \frac{\partial \hat{w}_y}{\partial \hat{y}}\bigg|_{\hat{y}=y} = \left( \frac{\hat{n}'_y}{n_y} \right)\bigg|_{\hat{y}=y} = -\frac{1}{y} \tag{2.14}$$

Again the outcome is clear. For male offspring, selection is always negative above minimum size $\delta$, and hence they are driven towards this minimum by selection. For queens, a candidate ESS is found at

$$-\frac{1}{x} + \frac{\beta}{x^2} = 0, \text{ or } x = \beta. \tag{2.15}$$

Hence, for model 2, as long as $\beta > \delta$, the queen candidate ESS is larger than male candidate ESS. A stability analysis again confirms the male and queen equilibria are both evolutionarily and convergence stable (electronic supplementary material, appendix).

## (c) Model 3

Now we consider a situation that does not have an analogue in the gamete evolution models of Bulmer & Parker [17]: multiple queens founding a colony. We modify model 1 (above) for this purpose. The $g$-functions are not altered, but consider the $s$-function. If colony survival depends on foundress resources as $s(z)$, then in this case $z$ will be composed of resources from multiple queens. Let us for simplicity assume that $s$ depends simply on the sum of resources from all founding queens, and the functional relationship $s(z)$ remains as in table 1. Hence, if $k$ queens in total contribute to a colony, of which one is a mutant, then colony survival is

$$s(z) = s(\hat{x} + (k-1)x) \tag{2.16}$$

where $\hat{x}$ is mutant queen size and $x$ resident queen size. Total mutant fitness via queen offspring is then

$$\hat{w}_x = \frac{1}{k} \hat{n}_x g(\hat{x}) s((\hat{x} + (k-1)x)). \tag{2.17}$$

Where we have assumed that on average colony fitness is split evenly between foundresses. This could correspond to, for example, equal sharing of the colony, or to only a single, randomly picked foundress surviving to ultimately reap all the benefits of the collaboratively founded colony, as usually is the case [25]. We therefore assume that although the resources an individual queen brings do alter survival prospects of the colony, they do not alter her expected relative share of fitness from a successful colony.

Total fitness via mutant sons

$$\hat{w}_y = \hat{n}_y g(\hat{y}) \frac{(1/k) n_x g(x) s(kx)}{n_y g(y)}. \tag{2.18}$$

The candidate ESS for male offspring size is as in model 1, but for queen size, the direction of selection is now determined by

$$\frac{1}{w_x} \frac{\partial \hat{w}_x}{\partial \hat{x}}\bigg|_{\hat{x}=x} = \left( \frac{\hat{n}'_x}{n_x} + \frac{g'(\hat{x})}{g(x)} + \frac{s'(\hat{x} + (k-1)x)}{s(kx)} \right)\bigg|_{\hat{x}=x}$$
$$= -\frac{1}{x} + \frac{\alpha}{x^2} + \frac{\beta}{k^2 x^2}, \tag{2.19}$$

and the candidate ESS is correspondingly

$$x = \alpha + \frac{\beta}{k^2}. \tag{2.20}$$

So, under the multiple foundress model, queens are still expected to be larger than males, but the size difference rapidly decreases with the number of queens. When there is only one queen ($k = 1$), model 3 is identical to model 1. Again, the results are convergence stable and evolutionarily stable when $k = 1$ or $k = 2$. For larger values of $k$, stability analysis is more subtle, but for a very wide range of biologically realistic parameter values, the equilibria remain stable (details are given in the electronic supplementary material, appendix).

## 4. Discussion

Our analysis raises a number of interesting points and may clarify understanding of both the evolution of queen–male dimorphism and that of anisogamy. First and foremost, we have confirmed the verbal idea [10] that the evolution of queen–male dimorphism can, at least in principle, evolve via similar logic as gamete dimorphism in classic models of the evolution of anisogamy (e.g. [12,17]). Compared with solitary lifestyles, the superorganism life cycle includes the additional stage of colony founding, which in turn introduces a new pressure: all fitness of the queen is dependent on survival of the colony that she founds until the stage where new queens and males are produced, and this colony-level reproduction is impossible before the 'somatic' colony has reached a certain size, such as a threshold number of workers [32]. By implication, if colony survival at the founding stage is dependent on the size of the queen, there will be a period of selection on queen size that is absent in the life cycle of a solitary insect. This is the reason we can 'frame-shift' upwards, and draw an analogy between queens, males and the superorganismal colony on the one hand, and sperm, ova and the organismal

zygote on the other. Such an analogy does not exist in solitary insects, because there is no equivalent of the zygote if we introduce a similar frame-shift (figure 1).

## (a) Differences between anisogamy evolution and sexual dimorphism in superorganisms

Despite the apparent analogy, we have also clarified a fundamental difference between anisogamy evolution and the 'superorganismal anisogamy' modelled here. In the current model, although male fitness is dependent on the survival of the colony, males do not contribute in any material way to the colony: they contribute only their genes, no matter what their size, in contrast with the evolution of gamete sizes where each gamete contributes resources according to size. This assumes that no resources that facilitate productivity are transmitted in the seminal fluid, which is currently unknown in social insects. Males therefore have no influence on the survival prospects of the colony. This is in stark contrast to models of the evolution of anisogamy, where two gametes fuse to combine not only genetic contributions, but their material contributions too (particularly when the two gametes are similar in size). It is this combination of material resources and the gametes' mutual influence on the survival prospects of the zygote that make the anisogamy model fundamentally coevolutionary in nature. It is also the reason why the necessary conditions for the divergence of gamete sizes in the anisogamy model are more stringent than the conditions for divergence of queen and male sizes in the current model. In the anisogamy model, the energy requirements of zygotes must be sufficiently larger than those of gametes to allow the divergence of gamete sizes [17]. In the current model, there is no such requirement. Even if colony survival places a fairly minor additional burden on the queen, queen–male dimorphism is expected to evolve.

A second difference is that in some superorganismal species, the colony is founded by more than one queen, whereas a zygote is founded by one ovum only: we have investigated the implications of this difference in model 3. Third, an ovum is fertilized by one sperm only (polyspermy typically leads to abnormal development [33,34]), and conversely, a spermatozoon fertilizes only one ovum. In superorganisms, neither is necessarily true. A queen can (and often does [35]) mate with multiple males, and a single male can potentially mate with several queens [20]. Neither of these differences alter our model: if mating is random and if all queens are fertilized, the *average* fitness gains per male remain the same regardless of these differences. Finally, in contrast with a female gamete of a normal organism which ceases to exist as an independent entity from the zygote stage onwards, the social insect queen remains a physically independent unit throughout the colony life, so the link between queen size and colony reproductive output is not necessarily completely severed. This would potentially further increase dimorphism, but the effect should be similar or smaller in superorganismal species relative to solitary species, because the relationship between queen size and reproductive output is decoupled to some extent by 'somatic' workers.

## (b) Differences between superorganisms and organisms

The key differences between the fitness effects of solitary female and superorganismal queens require some clarification. Selection for larger female size may, of course, well occur in a solitary species as well, both due to size-related increase in survival until mating, in further survival until reproduction, and fecundity. Thus, it might be challenging to tease apart whether precisely the 'anisogamy-like' processes would be driving queen–male dimorphism in social insects (if such a pattern does indeed hold up to systematic scrutiny of broad, phylogenetically controlled patterns). However, it is worth pointing out that (i) a size-related increase in survival until mating is unlikely to be systematically female-biased, (ii) size-related increase in survival between mating and reproduction is unlikely to apply in a majority of solitary species (i.e. outside species that do not construct nests, gather resources, defend a territory etc.). Furthermore, it is possible that large size provides fecundity benefits to a superorganismal queen as well, although indirectly, as the reproductive output is mediated by the 'somatic' worker phenotype of the colony. The key point remains: the obligatory growth stage of worker rearing after mating is a stage where positive selection for size occurs in queens but not males of superorganisms, but an analogous stage is lacking, or at least is not always present in solitary insects.

## (c) Testable predictions arising from the model

The model suggests several comparative tests [10]. The primary prediction is that dimorphism should be higher in superorganismal than solitary species, or species with more primitively social lifestyles. Second, model 3 shows that superorganismal species where multiple queens found a colony are expected to have lower size dimorphism than those where colonies have only a single foundress, and dimorphism is expected to decrease with the number of foundresses. Equation (2.20) suggests that size dimorphism should decrease quite quickly with the number of foundresses (i.e. in inverse proportion to the square of the foundress number, although the exact mathematical form will depend on model assumptions). The reason for the fast decrease is that each queen is able to gain from the resources brought by other queens, which allows 'exploitation' of the shared resource with a reduced contribution of one's own. These pattern may change, however, if queen size increases her chances of being the sole survivor until colony maturity (e.g. [36]).

Third, in species where the assumptions of the model are not met, dimorphism should be smaller. For example, effects of male size on mating or mate finding success may vary according to male life histories [37]. Selection pressures on queen size may also have been relaxed in species where workers from the mother colony accompany the foundress queen in so-called dependent colony founding through, for example, swarming or foundation of bud nests [38]. Dependent colony founding is the prevalent mode in *Apis* bees, and has evolved repeatedly in ants as well. Also differences in colony founding behaviour of single founding queens may introduce variation into selection for size. The benefits of size might be larger in queen in the so-called claustral founding species where the queen does not forage, but depends solely on her internal resources for rearing the first workers. Finally, biases in dispersal abilities of queens and males, and more generally lack of panmixia may affect selection for size (see [26] for a model of anisogamy evolution in a population where gamete dispersal is limited). While social insects with large-scale mating flights are often assumed to be panmictic [10], male-biased dispersal and limited gene flow through females is

also occasionally reported, especially in ants [39]. However, some degree of male-biased dispersal by itself does not necessarily breach the assumptions of our model: the central assumption is that males disperse to a sufficient extent that competition between sons of a rare mutant queen is negligible.

Fourth, while our model is based on a social Hymenoptera life cycle where the male dies soon after mating and does not contribute to colony life, an interesting comparison is provided by termites where both sexes stay in the colonies and contribute to the colony foundation and brood care. Thus, termites are more similar to the original anisogamy model, so the conditions for divergent selection are potentially more stringent and depending on a coevolutionary process. It should be noted, however, that a termite-specific model would still not be completely symmetrical as anisogamy models are: even if males contribute to colony founding, their contribution is of a fundamentally different kind than that of queens who are able to reproduce. There are reported cases where termite queens are slightly larger than the males [40,41], but broad comparisons across species are lacking. A further complication for testing the predictions may arise from selection for larger males due to male–male competition [42].

Models of anisogamy evolution under gamete competition make testable predictions, some of which apply to the current model. A prediction that is transferrable to the current model is that gamete dimorphism is predicted to increase with increasing size and complexity of the adult organism [12,17,43], and this prediction has some empirical support from comparative studies [44–46]. In the superorganism model, the analogous prediction is that queen–male dimorphism is predicted to increase with increasing size (i.e. number of workers in a colony at sexual maturity) and complexity of the colony. While colony complexity might be difficult to quantify unambiguously, reasonable proxies include, for example, the presence and number of worker subcastes, or number of workers. In the model, the reason for this prediction is that a larger and more complex superorganism probably requires a larger amount of initial resources for reasonable chances of survival, so that we should expect a positive correlation between size/complexity and the parameter $\beta$ in the model. This assumed correlation again goes back to an analogy with models of anisogamy evolution. In the gamete competition model of anisogamy evolution, the parameter $\beta$ in the zygote

survival function $s$ increases with the need to provision the embryo, and the needs of the embryo are in turn thought to increase with increasing multicellular complexity of the adult organism [17]. Analogously, the parameter $\beta$ in the colony survival function $s$ in the current model is thought to correlate with the complexity of the colony.

A prediction that is *not* transferrable is that anisogamy should only evolve if there is a sufficiently large difference between the minimum requirements of the gamete and zygote [17], which also suggests that there should typically be a near-stepwise distribution from isogamy with small gametes to anisogamy with a relatively large difference in size between the male and female gamete [47] (this prediction has some empirical support [47]). This prediction is not replicated in the superorganism model because the coevolutionary link between queen and male size is broken (see model 1 description). Hence, the current models predict a relatively continuous distribution of queen–male dimorphism in superorganisms, compared to the expected stepwise distribution of gamete dimorphism from isogamy to anisogamy.

To conclude, we have shown how adjusting a classic model of anisogamy evolution to the biological particularities of social insect colonies as superorganisms can elucidate the evolution of size dimorphism among social insect queens and males. Thus, the superorganism metaphor can be used as a starting point to derive testable predictions, and ultimately increase our understanding of insect societies, building on theoretical and empirical understanding of how multicellular organisms evolve. Furthermore, the model highlights similarities and crucial differences of evolutionary processes at different hierarchical levels, and thus increases our understanding of the major transitions in evolution.

Data accessibility. This article has no additional data.

Authors' contributions. Both authors contributed to all aspects of the manuscript.

Competing interests. We declare we have no competing interests.

Funding. J.L. is funded by an Australian Research Council Discovery Early Career Research Award (project no. DE180100526) from the Australian Government. H.H. has been funded by the Kone Foundation.

Acknowledgements. We are grateful to Jacobus J. Boomsma, Madeleine Beekman, Geoff A. Parker and two anonymous referees for helpful comments on earlier drafts of the manuscript.

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
