## [Reviewer comments · Proceedings of the Royal Society B: Biological Sciences]

Review History

RSPB-2020-0635.R0 (Original submission)

Review form: Reviewer 1 (Kazuki Tsuji)

Recommendation

Accept with minor revision (please list in comments)

Scientific importance: Is the manuscript an original and important contribution to its field?

Acceptable

General interest: Is the paper of sufficient general interest?

Acceptable

Quality of the paper: Is the overall quality of the paper suitable?

Good

Is the length of the paper justified?

Yes

Should the paper be seen by a specialist statistical reviewer?

No

Do you have any concerns about statistical analyses in this paper? If so, please specify them explicitly in your report.

No

It is a condition of publication that authors make their supporting data, code and materials available - either as supplementary material or hosted in an external repository. Please rate, if applicable, the supporting data on the following criteria.

Is it accessible?

Yes

Is it clear?

Yes

Is it adequate?

Yes

Do you have any ethical concerns with this paper?

No

Comments to the Author

Rigorous comparison between organisms and superorganisms is a less-explored but important issue in evolutionary biology. This manuscript addressed the question if the gamete competition theory in the organism-level for the evolution of anisogamy can be extended to explain the superorganism level phenomenon, namely sexual dimorphism of reproductive castes in social insects. The authors also discussed what conditions specific to super organisms would violate assumptions of organism-level theory. This way of question raising is sensible. Mathematical modellings following the Bulmer and Parker's approach are simple enough and clear, and calculations seemed to be all correct as far as I see. I have only a few comments summarized below. I hope they would be useful.

What is lacking in the ms. is the consideration on the relationship between size and dispersal distance. In the evolution of anisogamy the size-dispersal tradeoff, i.e. eggs are less able to disperse than sperms, has been discussed. A similar difference may exist in sexual offspring of social insects. It is worth mentioning that the assumption of a panmictic population may holds, even in the presence of a certain degree of sexual difference in the dispersal distance.

Lines 371-375. As the authors correctly pointed out in termites both the king and the queen contribute resources for their colony's survival. Therefore, sexual dimorphism is predicted to be maximized through game processes. There are empirical studies showing that termite alates females are larger than males, for example (Matsuura 2005 *Ann. Entomol. Soc. Am.* 99: 625-628) . However, in my personal experiences the sexual dimorphism is less evident in many termites, because alate sexing is often a difficult job due to similar sized males and females. I guess, the stringent monogamy at the colony-founding stage of termites may be something do to with this. A non-affle type male-male competition may commonly exist in termites (see Matsuura et al. 2002, *J. Theor. Biol.* 214: 63-70).

The Alpha and the beta are key parameters that determine individual fitness of male and female offspring in a given body size. More precise explanations of the non-linear relationship will help readers' understanding. For instance, the increase of $g(z)$ per an unit increase of alpha differs when $\alpha > z$ and when $\alpha < z$. What does this imply in real organisms?

Lines 377-379. The above issue must be related also to the phrase "prediction that is transferable to the current model is that gamete dimorphism is predicted to increase with increasing size and complexity of the adult organism [12, 17, 35], and this prediction has some empirical support from comparative studies [36-38]." However, I am not sure what parameter and formula in the

manuscript describe “increase complexity of the adult organism”. Please clarify.

Review form: Reviewer 2

Recommendation

Accept as is

Scientific importance: Is the manuscript an original and important contribution to its field?

Good

General interest: Is the paper of sufficient general interest?

Good

Quality of the paper: Is the overall quality of the paper suitable?

Excellent

Is the length of the paper justified?

Yes

Should the paper be seen by a specialist statistical reviewer?

No

Do you have any concerns about statistical analyses in this paper? If so, please specify them explicitly in your report.

No

It is a condition of publication that authors make their supporting data, code and materials available - either as supplementary material or hosted in an external repository. Please rate, if applicable, the supporting data on the following criteria.

Is it accessible?

N/A

Is it clear?

N/A

Is it adequate?

N/A

Do you have any ethical concerns with this paper?

No

Comments to the Author

This paper explores an analogy that has been proposed between the evolution of gamete size dimorphism in multicellular organisms and the evolution of male-female size dimorphism in social insects, with males and females being essentially gametes of the superorganismal colony. Mathematical models of male-female dimorphism are constructed that follow, as far as seems reasonable, previous models of gamete size dimorphism. There are a few differences in assumptions, guided by the different biologies for the two cases. As a result, the models do show some correspondence in the evolution of the two phenomena, but also some differences. The authors are quite clear in pointing out what these differences are and why they emerge from the differences in biological assumptions.

The authors also provide some testable predictions of their model, though they do not do such test themselves. Such tests would require considerable comparative data and careful phylogenetic analysis and should not be expected in this initial paper outlining the theory.

There are some assumptions made here that would be questionable for some cases. For example, the assumption that males are in a lottery for females in which male size does not matter could be questioned (particularly when queens mate multiply and sperm volume might matter for paternity). But I think the assumption is OK for the purposes of exploring the potential deep similarities with gamete size evolution. All models make simplifying assumptions.

The multiple-queen model shows that less size dimorphism is expected. It could be pointed out that this is basically because of a tragedy of the commons among the queens, with each being able to gain from the resources brought by the others.

equation 5. If you want to maximize your audience, give a little more explanation of the logic here, or at least a reference. Theoreticians will of course know exactly what you are doing, but maybe not some others.

Decision letter (RSPB-2020-0635.R0)

03-Apr-2020

Dear Dr Lehtonen:

Your manuscript has now been peer reviewed and the reviews have been assessed by an Associate Editor. The reviewers' comments (not including confidential comments to the Editor) and the comments from the Associate Editor are included at the end of this email for your reference. As you will see, the reviewers and the Editors have raised some concerns with your manuscript and we would like to invite you to revise your manuscript to address them.

Research ethics:

Use of animals and field studies:

Please submit a copy of your revised paper within three weeks. If we do not hear from you within this time your manuscript will be rejected. If you are unable to meet this deadline please let us know as soon as possible, as we may be able to grant a short extension.

Best wishes,
Dr Sasha Dall
mailto: proceedingsb@royalsociety.org

Associate Editor

Comments to Author:

Your manuscript has now been seen by two experts in evolutionary modeling, and I am happy to say that they see real value in your work. Both have made suggestions for small additions and clarifications that need to be made to the manuscript before it is published, and I hope that you are able to make these as requested. Please detail responses and revisions with what I hope will be the final version of your manuscript, which I look forward to seeing.

Reviewer(s)' Comments to Author:

Referee: 1

Comments to the Author(s)

Rigorous comparison between organisms and superorganisms is a less-explored but important issue in evolutionary biology. This manuscript addressed the question if the gamete competition theory in the organism-level for the evolution of anisogamy can be extended to explain the superorganism level phenomenon, namely sexual dimorphism of reproductive castes in social insects. The authors also discussed what conditions specific to super organisms would violate assumptions of organism-level theory. This way of question raising is sensible. Mathematical modellings following the Bulmer and Parker's approach are simple enough and clear, and calculations seemed to be all correct as far as I see. I have only a few comments summarized below. I hope they would be useful.

What is lacking in the ms. is the consideration on the relationship between size and dispersal distance. In the evolution of anisogamy the size-dispersal tradeoff, i.e. eggs are less able to disperse than sperms, has been discussed. A similar difference may exist in sexual offspring of social insects. It is worth mentioning that the assumption of a panmictic population may holds, even in the presence of a certain degree of sexual difference in the dispersal distance.

Lines 371-375. As the authors correctly pointed out in termites both the king and the queen contribute resources for their colony's survival. Therefore, sexual dimorphism is predicted to be maximized through game processes. There are empirical studies showing that termite alates females are larger than males, for example (Matsuura 2005 *Ann. Entomol. Soc. Am.* 99: 625-628). However, in my personal experiences the sexual dimorphism is less evident in many termites, because alate sexing is often a difficult job due to similar sized males and females. I guess, the stringent monogamy at the colony-founding stage of termites may be something do to with this. A non-affle type male-male competition may commonly exist in termites (see Matsuura et al. 2002, *J. Theor. Biol.* 214: 63-70).

The Alpha and the beta are key parameters that determine individual fitness of male and female offspring in a given body size. More precise explanations of the non-linear relationship will help readers' understanding. For instance, the increase of $g(z)$ per an unit increase of alpha differs when $\alpha > z$ and when $\alpha < z$. What does this imply in real organisms?

Lines 377-379. The above issue must be related also to the phrase "prediction that is transferable to the current model is that gamete dimorphism is predicted to increase with increasing size and complexity of the adult organism [12, 17, 35], and this prediction has some empirical support

from comparative studies [36-38].” However, I am not sure what parameter and formula in the manuscript describe “increase complexity of the adult organism”. Please clarify.

Referee: 2

Comments to the Author(s)

This paper explores an analogy that has been proposed between the evolution of gamete size dimorphism in multicellular organisms and the evolution of male-female size dimorphism in social insects, with males and females being essentially gametes of the superorganismal colony. Mathematical models of male-female dimorphism are constructed that follow, as far as seems reasonable, previous models of gamete size dimorphism. There are a few differences in assumptions, guided by the different biologies for the two cases. As a result, the models do show some correspondence in the evolution of the two phenomena, but also some differences. The authors are quite clear in pointing out what these differences are and why they emerge from the differences in biological assumptions.

The authors also provide some testable predictions of their model, though they do not do such test themselves. Such tests would require considerable comparative data and careful phylogenetic analysis and should not be expected in this initial paper outlining the theory.

There are some assumptions made here that would be questionable for some cases. For example, the assumption that males are in a lottery for females in which male size does not matter could be questioned (particularly when queens mate multiply and sperm volume might matter for paternity). But I think the assumption is OK for the purposes of exploring the potential deep similarities with gamete size evolution. All models make simplifying assumptions.

The multiple-queen model shows that less size dimorphism is expected. It could be pointed out that this is basically because of a tragedy of the commons among the queens, with each being able to gain from the resources brought by the others.

equation 5. If you want to maximize your audience, give a little more explanation of the logic here, or at least a reference. Theoreticians will of course know exactly what you are doing, but maybe not some others.

Author's Response to Decision Letter for (RSPB-2020-0635.R0)

See Appendix A.

Decision letter (RSPB-2020-0635.R1)

17-Apr-2020

Dear Dr Lehtonen

I am pleased to inform you that your manuscript entitled "Superorganismal anisogamy: Queen-male dimorphism in eusocial insects" has been accepted for publication in Proceedings B.

You can expect to receive a proof of your article from our Production office in due course, please check your spam filter if you do not receive it. PLEASE NOTE: you will be given the exact page

length of your paper which may be different from the estimation from Editorial and you may be asked to reduce your paper if it goes over the 10 page limit.

Open Access

Your article has been estimated as being 9 pages long. Our Production Office will be able to confirm the exact length at proof stage.

Paper charges

Sincerely,

Dr Sasha Dall

Associate Editor:

Board Member

Comments to Author:

Thanks for your positive and constructive response to the changes requested. I am happy to recommend final acceptance of your manuscript, and look forward to seeing it when it is published

03-Apr-2020

Dear Dr Lehtonen:

Your manuscript has now been peer reviewed and the reviews have been assessed by an Associate Editor. The reviewers' comments (not including confidential comments to the Editor) and the comments from the Associate Editor are included at the end of this email for your reference. As you will see, the reviewers and the Editors have raised some concerns with your manuscript and we would like to invite you to revise your manuscript to address them.

Research ethics:

Use of animals and field studies:

Please submit a copy of your revised paper within three weeks. If we do not hear from you within this time your manuscript will be rejected. If you are unable to meet this deadline please let us know as soon as possible, as we may be able to grant a short extension.

Best wishes,

Dr Sasha Dall
proceedingsb@royalsociety.org

Associate Editor

Comments to Author:

Your manuscript has now been seen by two experts in evolutionary modeling, and I am happy to say that they see real value in your work. Both have made suggestions for small additions and clarifications that need to be made to the manuscript before it is published, and I hope that you are able to make these as requested. Please detail responses and revisions with what I hope will be the final version of your manuscript, which I look forward to seeing.

Dear Editors,

Many thanks for your kind comments, and for the opportunity to submit a revised version of our manuscript. We have now addressed all of the reviewers' concerns and have detailed our responses to them below. Our responses are written in red font.

In this same file, immediately following our responses to the reviewers' queries you will also find a 'track changes' version of our revised manuscript, indicating all changes made since the first submission.

We hope our revisions are satisfactory, and look forward to hearing your decision.

Sincerely,
Jussi Lehtonen
Heikki Helanterä

Reviewer(s)' Comments to Author:

Referee: 1

Comments to the Author(s)

Rigorous comparison between organisms and superorganisms is a less-explored but important issue in evolutionary biology. This manuscript addressed the question if the gamete competition theory in the organism-level for the evolution of anisogamy can be extended to explain the superorganism level phenomenon, namely sexual dimorphism of reproductive castes in social insects. The authors also discussed what conditions specific to super organisms would violate assumptions of organism-level theory. This way of question raising is sensible. Mathematical modellings following the Bulmer and Parker's approach are simple enough and clear, and calculations seemed to be all correct as far as I see. I have only a few comments summarized below. I hope they would be useful.

What is lacking in the ms. is the consideration on the relationship between size and dispersal distance. In the evolution of anisogamy the size-dispersal tradeoff, i.e. eggs are less able to disperse than sperms, has been

organism [17]. Analogously, the parameter β in the colony survival function s in the current model is thought to correlate with the complexity of the colony.”

Referee: 2

Comments to the Author(s)

This paper explores an analogy that has been proposed between the evolution of gamete size dimorphism in multicellular organisms and the evolution of male-female size dimorphism in social insects, with males and females being essentially gametes of the superorganismal colony. Mathematical models of male-female dimorphism are constructed that follow, as far as seems reasonable, previous models of gamete size dimorphism. There are a few differences in assumptions, guided by the different biologies for the two cases. As a result, the models do show some correspondence in the evolution of the two phenomena, but also some differences. The authors are quite clear in pointing out what these differences are and why they emerge from the differences in biological assumptions.

The authors also provide some testable predictions of their model, though they do not do such tests themselves. Such tests would require considerable comparative data and careful phylogenetic analysis and should not be expected in this initial paper outlining the theory.

There are some assumptions made here that would be questionable for some cases. For example, the assumption that males are in a lottery for females in which male size does not matter could be questioned (particularly when queens mate multiply and sperm volume might matter for paternity). But I think the assumption is OK for the purposes of exploring the potential deep similarities with gamete size evolution. All models make simplifying assumptions.

The multiple-queen model shows that less size dimorphism is expected. It could be pointed out that this is basically because of a tragedy of the commons among the queens, with each being able to gain from the resources brought by the others.

This is a good point, and we have added an explanation (p.17-18):

“Equation 20 suggests that size dimorphism should decrease quite quickly with the number of foundresses (i.e. in inverse proportion to the square of the foundress number). The reason for the fast decrease is that each queen is able to gain from the resources brought by other queens, which allows ‘exploitation’ of the shared resource with a reduced contribution of one’s own.”

We feel the tragedy of the commons may be slightly out of place here, and we would prefer to not mention it explicitly. Rankin et al (The tragedy of the commons in evolutionary biology, Daniel J. Rankin, Katja Bargum and Hanna Kokko, TRENDS in Ecology and Evolution Vol.22 No.12) define the tragedy of the commons as a “situation in which individual competition reduces the resource over which individuals compete, resulting in lower overall fitness for all members of a group or population.”. While there is a similarity to the current model, in our model the individuals contribute the resource themselves, which is typically not the case in a tragedy of the commons scenario (e.g. over-exploitation of fish stocks).

equation 5. If you want to maximize your audience, give a little more explanation of the logic here, or at least a reference. Theoreticians will of course know exactly what you are doing, but maybe not some others.

We agree, and have included the following explanation after equations 5-6:

“The method used in equations 5-6 is that of evolutionary game theory [29], which is in the context of this model effectively equivalent to adaptive dynamics (i.e. the equations are of similar form in both methodologies; see [30]). In words, equations 5-6 amount to estimating selection on mutants that deviate from the resident size by a small amount.”

And:

“The candidate evolutionarily stable strategies (x^* and y^*) for male and queen size can be solved by setting both equations equal to zero and solving for x and y (i.e. finding values for queen and male sizes where selection vanishes).”

Although not explicitly requested, we have also added a short explanation of checking for evolutionary and convergence stability to the beginning of the appendix, along with relevant references.

Journal Name: Proceedings of the Royal Society B

Journal Code: RSPB

Print ISSN: 0962-8452

Online ISSN: 1471-2954

Journal Admin Email: proceedingsb@royalsociety.org

MS Reference Number: RSPB-2020-0635

Article Status: SUBMITTED

MS Dryad ID: RSPB-2020-0635

MS Title: Superorganismal anisogamy: Male-queen dimorphism in eusocial insects

MS Authors: Lehtonen, Jussi; Helanterä, Heikki

Contact Author: Jussi Lehtonen

Contact Author Email: jussi.lehtonen@iki.fi

Contact Author Address 1:

Contact Author Address 2:

Contact Author Address 3:

Contact Author City: Sydney

Contact Author State: New South Wales

Contact Author Country: Australia

Contact Author ZIP/Postal Code: 2006

Keywords: Anisogamy, size dimorphism, queen, male, superorganism

Abstract: Colonies of insects such as ants and honeybees are commonly viewed as 'superorganisms', with division of labour between reproductive "germline-like" queens and males and "somatic" workers. On this view, properties of the superorganismal colony are comparable to those of solitary organisms to such an extent that the colony itself can be viewed as a unit analogous to an organism. Thus the concept of a superorganism can be useful as a guide to thinking

about life-history and allocation traits of colonies as a whole. A pattern that seems to reoccur in insects with superorganismal societies is size dimorphism between queens and males, where queens tend to be larger than males. It has been proposed that this is analogous to the phenomenon of anisogamy at the level of gametes in organisms with separate sexes; more specifically, it is suggested that this caste dimorphism may have evolved via similar selection pressures as gamete dimorphism arises in the 'gamete competition' theory for the evolution of anisogamy. In this analogy, queens are analogous to female gametes, males are analogous to male gametes, and colony survival is analogous to zygote survival in gamete competition theory. Here we explore if this question can be taken beyond an analogy, and whether a mathematical model at the superorganism level, analogous to gamete competition at the organism level, may explain the caste dimorphism seen in superorganismal insects. We find that the central theoretical idea holds, but that there are also significant differences between the way this generalised 'propagule competition' theory operates at the levels of solitary organisms and superorganisms. In particular, we find that the theory can explain superorganismal caste dimorphism, but compared to anisogamy evolution a central coevolutionary link is broken, making the requirements for the theory to work less stringent from those found for the evolution of anisogamy. EndDryadContent